# DNMT3B Is an Oxygen-Sensitive De Novo Methylase in Human Mesenchymal Stem Cells

**DOI:** 10.3390/cells10051032

**Published:** 2021-04-27

**Authors:** Fatma Dogan, Rakad M Kh Aljumaily, Mark Kitchen, Nicholas R. Forsyth

**Affiliations:** 1The Guy Hilton Research Laboratories, Faculty of Medicine and Health Sciences, School of Pharmacy and Bioengineering, Keele University, Stoke on Trent ST5 5BG, UK; f.dogan@keele.ac.uk (F.D.); m.o.kitchen@keele.ac.uk (M.K.); 2Department of Biology, College of Science, University of Baghdad, Baghdad 17635, Iraq; rakad.aljumaily@sc.uobaghdad.edu.iq

**Keywords:** mesenchymal stem cells, characterisation, epigenetic, methylation, hydroxymethylation, physiological oxygen, DNA methyltransferase

## Abstract

The application of physiological oxygen (physoxia) concentrations is becoming increasingly commonplace within a mammalian stem cell culture. Human mesenchymal stem cells (hMSCs) attract widespread interest for clinical application due to their unique immunomodulatory, multi-lineage potential, and regenerative capacities. Descriptions of the impact of physoxia on global DNA methylation patterns in hMSCs and the activity of enzymatic machinery responsible for its regulation remain limited. Human bone marrow-derived mesenchymal stem cells (BM-hMSCs, passage 1) isolated in reduced oxygen conditions displayed an upregulation of SOX2 in reduced oxygen conditions vs. air oxygen (21% O_2,_ AO), while no change was noted for either OCT-4 or NANOG. DNA methylation marks 5-methylcytosine (5mC) and 5-hydroxymethylcytosine (5hmC) showed decreases in 2% O_2_ environment (workstation) (2% WKS). DNMT3B (DNA methyltransferase 3B) and TET1 (Ten-eleven translocation enzyme 1) displayed reduced transcription in physoxia. Consistent with transcriptional downregulation, we noted increased promoter methylation levels of DNMT3B in 2% WKS accompanied by reduced DNMT3B and TET1 protein expression. Finally, a decrease in HIF1A (Hypoxia-inducible factor 1A) gene expression in 2% WKS environment correlated with protein levels, while HIF2A was significantly higher in physoxia correlated with protein expression levels vs. AO. Together, these data have demonstrated, for the first time, that global 5mC, 5hmC, and DNMT3B are oxygen-sensitive in hMSCs. Further insights into the appropriate epigenetic regulation within hMSCs may enable increased safety and efficacy development within the therapeutic ambitions.

## 1. Introduction

MSCs, discovered by Friedenstein et al., and coined by Caplan, are non-hematopoietic bone marrow-derived stem cells with the capacity to undergo multipotent differentiation into a range of mesodermal tissues [1,2]. hMSCs therapeutic applications span widespread clinical trial through to market-place products across a range of clinical applications, including ischaemic stroke, diabetes, and myocardial infarction in addition to musculoskeletal tissue repair [3]. A key feature of hMSCs that promotes their clinical utility is their overall lack of immunogenicity permitting safe allogeneic cell transplantation without the need for immunosuppression [4]. The precise role of hMSCs (early passages 1–5 times, bone marrow) in repair stimulation remains unclear, but evidence continues to emerge that this may be due to a combination of paracrine activity, transfer of exosomes or microvesicles, and differentiation into replacement cell types via their inherent differentiation capacity [5].

Epigenetics describes the heritable change in gene function without an accompanying change in DNA sequence. Epigenetic changes can modulate gene expression and can result from environmental influences, including those encountered in routine mammalian cell culture, i.e., air oxygen. Epigenetic mechanisms with the potential to modulate gene expression include DNA methylation, histone modification, and non-coding RNAs. These play a critical role during development and differentiation [6,7]. DNA methylation is associated with stem cell differentiation, stem cell renewal, and also replicative senescence [8,9]. De novo methyltransferase enzymes (DNMT3A and DNMT3B) are highly expressed in undifferentiated stem cells during development, while subsequent expression during the onset of differentiation is described as being either reduced [10,11], or alternatively, that DNMT3A is increased and DNMT3B decreased during differentiation [12]. The TET family of proteins catalyse the oxidation of 5mC to 5hmC and levels of 5hmC should also be taken into account when considering DNA methylation [13]. 5hmC levels are noted as high in mouse embryonic stem cells (mESCs) and human embryonic stem cells (hESCs) [14], and TET1 is recognised as important for mESC via maintenance of Nanog expression and playing a role in the inner cell mass cell specification [15]. The mammalian embryo develops under physoxia where HIF factors physically interact with histone deacetylase enzymes (HDAC) [16] and removal of methyl groups depends on oxygenases [17]. These emphasise the key role of oxygen in determining epigenetic profiles. Physiological oxygen levels drive a significant reduction of global DNA methylation levels across a range of tumour-derived cell lines, including those derived from colorectal and melanoma cancers [18]. Similarly, low oxygen conditions influence genome-wide DNA hydroxymethylation and TET1 expression levels in a HIF-dependent manner [19]. For example, TET1, TET3, and 5hmC levels are correlated with tumour hypoxia and malignancy in breast cancer patients, while hypoxia is tightly associated with high-grade breast tumour [20]. TET1 and TET3 associate with the activation of TNFα–p38–MAPK signalling as a response to hypoxia, where hydroxymethylation of TNFα results in the upregulation of gene expression and downstream pathways [20]. A direct link between oxygen, differentiation, self-renewal, and methylation was established in hESC, where differentiated cells were reverted to a stem cell-like state via reduced oxygen exposure, resulting in promoter hypomethylation and reactivation of Oct4. Air-based culture and serum-forced differentiation of hESCs resulted in a highly methylated Oct4 promoter as compared to significantly lower levels in reduced oxygen cultured cells [21]. Taken together, these observations emphasise the effect of oxygen on methylation status in both cancer and stem cell populations.

Mesenchymal stem cells are localised within perivascular niches (microenvironment around a vessel) in mammals [22]. Physoxia levels within bone marrow sits within the 1–6% range [23] and the in vitro effect of these low oxygen concentrations on the maintenance of characteristics of MSCs, including their proliferation, differentiation, immune regulatory capacity, and multipotency, are well established in early passages 0–5 and late passages 10 [24,25,26]. The impact of reduced oxygen on tri-lineage differentiation of MSCs remains inconclusive, with some studies suggesting an increased differentiation potential in human MSCs urine stem cells (USCs), dental pulp stem cells (DPSCs), amniotic fluid stem cells (AFSCs), BM-MSCs (passage 5–15), and human umbilical cords (passage 9–11) [25,27], while others suggest decreased levels of osteogenic and adipogenic lineages [28,29]. The persistence of these inconsistencies may be due to the range of culture condition applied and the extent of variability in engineering measures applied to control oxygen levels [24,30]. Methylation changes at specific CpG sites (hyper- or hypomethylation) are also reported as affecting the differentiation of MSCs (adipose tissue-derived, passage 2–4) with correlated hypomethylated CpGs and transcription factor binding sites, including Sp1, TFIIB, C/EBPα, Pax8, and N-myc [31,32]. Furthermore, CpG sites within homeobox- and differentiation-related genes, including DLX5 (distal-less homeobox 5), HOXA5, C10orf27 (chromosome 10 open reading frame 27), RUNX3 (runt-related transcription factor 3), and CDKN2B (cyclin-dependent kinase inhibitor 2B), are differentially methylated in response to a long period of culture and aging of human mesenchymal stromal cells (passage 8–15) [8]. siRNA knockdown demonstrated that TET1 had an inhibitory effect on BM-hMSCs (derived from the posterior iliac crest) osteogenic and adipogenic differentiation, TET2 promoted it, and TET3 displayed no role [33]. Finally, BM-hMSCs undergoing osteogenic and adipogenic differentiation demonstrated increased levels of 5hmC that correlated with reduced 5mC levels [33]. 

The effects of reduced oxygen on DNA methylation in hMSCs have received little attention thus far. To our knowledge, we report the first investigation of epigenetic marks (5mC and 5hmC) alterations in response to physoxia in BM-hMSCs. We evaluated expression levels of enzymes linked to control of global DNA methylation; DNMT1 (DNA methyltransferase 1), DNMT3A, DNMT3B, DNMT3L, TET1 (Ten-eleven translocation enzyme 1), TET2, TET3, and the extent of their promoter-specific methylation. Expression of Hypoxia-Inducible Factor (HIF)1A and HIF2A were also determined across the three oxygen settings. BM-hMSCs cultured regularly in physoxia showed a significant decline in 5mC and 5hmC, followed by transcriptional and translational decrease in DNMT3B and TET1 with accompanying promoter-specific methylation alterations. In summary, the level of 5mC and 5hmC in BM-hMSCs are sensitive to oxygen tension with associated DNMT3B transcriptional and translational modulation. This research aims to investigate the effect of physioxa on global DNA methylation and the enzymes responsible for its maintenance in BM-hMSCs.

## 2. Materials and Methods

### 2.1. Isolation and Culture of BM-hMSCs

Three human BM-hMSCs BMA-16 (Male, Age—29 years, Lonza, MD, USA), BMA-20 (Female, Age—50 years, Lonza, MD, USA) and BMA-25 (Male, Age—27 years, AllCells, LLC, Alameda, CA, USA) were isolated from commercially sourced donor bone marrow and used experimentally at passage 1 according to previously described methodology [24]. Isolation and maintenance of hMSCs were in either standard 21% O_2_ cell culture (AO), 2% O_2_ deoxygenated media using HypoxyCool system (Baker Ruskinn, Bridgend, UK) and cultured in 2% O_2_ incubator (2%PG) with handling in a biological safety cabinet, or 2% O_2_-HypoxyCool media in a Sci-Tive workstation (Baker Ruskinn) (2%WKS). Oxygen defined media was obtained by deoxygenation to a 2% oxygen level using the present parameters within a HypoxyCool.

### 2.2. Flow Cytometric Analysis

Following passage 1 × 10^5^ cells were collected into 1.5 mL tubes, then centrifuged, and the supernatant removed. IgG1 (mouse monoclonal, 130-098-845, Miltenyi Biotec, Bergisch Gladbach, Germany), CD19 (130-098-168, Miltenyi Biotec), CD73 (130-097-943, Miltenyi Biotec), CD90 (130-097-932, Miltenyi Biotec), CD105 (130-098-906, Miltenyi Biotec), IgG2a (mouse monoclonal, 130-098-849, Miltenyi Biotec), CD14 (130-098-167, Miltenyi Biotec), CD34 (130-098-140, Miltenyi Biotec), CD45 (130-098-141, Miltenyi Biotec), and HLA-DR (130-098-177, Miltenyi Biotec) were diluted (1:1000) in FACS (PBS with 0.5% (*w*/*v*) BSA (Fisher Scientific, Loughborough, UK ) buffer and added to specific tubes for 15 min at 4 °C in the dark. After incubation, cells were washed with FACS buffer and 5 min centrifuged at 900 rpm. Pellets were then resuspended in 300 μL of FACS buffer. A total of 50,000 events were recorded for each sample using Beckman Coulter Cytomics FC 500.

### 2.3. Quantification of Global DNA Methylation (5mC and 5hmC) Levels

DNeasy Blood and Tissue kit (Qiagen, Manchester, UK) were used to isolate total genomic DNA. Quantification of total 5mC content was performed following instructions of MethylFlashTM Methylated DNA quantification Kit (Epigentek, Farmingdale, NJ, USA) using 100 ng of input DNA per reaction. Total 5hmC quantification was analysed with the MethylFlashTM Hydroxymethylated DNA Quantification Kit (Epigentek, USA) using 200 ng of input DNA per reaction. Manufacturer’s protocols were followed in both instances followed by colourimetric determination at A450 nm via a microplate reader (BioTek, Synergy 2, Gen5 1.10, Swindon, UK and conversion through standard curve analysis into global DNA methylation levels.

### 2.4. Gene Expression Analysis

RNA was extracted from hMSCs using the RNeasy^®^ Mini Kit (Qiagen) following the manufacturer’s protocol. Total RNA concentration was quantified using a NanoDropTM 2000/2000c Spectrophotometer (Thermo Scientific, Oxford, UK. QuantiFast SYBR^®^ Green RT-PCR Kit was used to perform reverse transcriptase polymerase chain reaction. PCR primers were designed using GenBank (http://www.ncbi.nlm.nih.gov/, accessed on 11 January 2017) and oligos ordered from Invitrogen Ltd. Primer sequences and product sizes are listed in Appendix A. Run parameters were 30 min of reverse transcriptase activation at 50 °C followed by activation at 95 °C for 15 min and denaturation at 95 °C for 1 min. This was followed by 39 cycles of denaturation at 95 °C, annealing at 55 °C, and extension at 72 °C for 1 min before a final elongation step at 72 °C for 10 min. The same programme setting was used for all primers. The only exception was DNMT3A, which had a 56 °C annealing temperature.

### 2.5. Protein Analysis

Following passage, cells were first washed with PBS before being lysed with fresh RIPA buffer and centrifuged at 10,000× *g* for 10 min at 4 °C. Total protein concentration in the supernatant was measured using the BCA protein kit (Sigma-Aldrich). Western Blot analysis was performed on 30 μg of protein using DNMT3B (R&D System/MAB7646, Secondary Anti-mouse IgG-HRP, Cell Signalling/70765, London, UK, TET1 (ThermoFisher/GT1462, Secondary Anti-mouse IgG-HRP, Cell Signalling/70765, London, UK, HIF1A (Novusbio/NB100-479, Secondary Anti-rabbit IgG-HRP Cell Signalling/70745, London, UK, HIF2A (Novusbio/NB100-122, Secondary Anti-rabbit IgG-HRP Cell Signalling/70745), and GAPDH (Merck/MAB374, Secondary Anti-mouse IgG-HRP, Cell Signalling/70765, London, UK). The immunoreactivity bands were subjected to UpLight HRP chemiluminescent substrate solution (Uptima) and imaged using a FluorChem M Imager.

### 2.6. Methylation Analysis

Bisulphite conversion of 500 ng input genomic DNA per sample was performed using EZ DNA Methylation-Gold™ Kit (Zymo Research, Orange, CA, USA). In brief, 500 ng gDNA was diluted in 20 μL nuclease free water and 130 μL of CT Conversion reagent were mixed, which was placed in a PCR tube at 98 °C for 10 min followed by 64 °C for 2.5 h. Then, M-Binding Buffer and M-Desulphonation Buffer treatment steps were followed by washing and elution as described. Converted DNA was measured using Nanodrop spectrophotometry (single stranded DNA) and then stored at −20 °C.

The CpG island sequence information of selected genes was obtained from UCSC genome browser (https://genome.ucsc.edu, accessed on 12 July 2016). Primers specific for DNMT1, DNMT3A, DNMT3B, DNMT3L, TET1, TET2, and ET3 were designed via the PyroMark Q24 Software 2.0, and designed primers were supplied by Biomers (Biomers, Ulm, Germany) (Appendix A). One primer within each pair was biotinylated at the 5′ end to enable PCR amplicon antisense strand sequencing. PCR amplification was performed with converted DNA (2–4 μL). Amplification of samples was performed with the GoTaq^®^G2 Flexi DNA Polymerase kit (Promega, Southampton, UK) in 25 μL volumes containing 2.5 μL of 25 mM MgCl2, 1 μL 20 μM/μL nucleotides, 1 μL of each forward and reverse primer, 5 μL 5× Flexi buffer, 0.2 μL Taq DNA Polymerase, and 12.3 μL molecular grade water. Cycling parameters were one cycle of 95 °C for 5 min for initial denaturation followed by touch-down PCR for the first 14 cycles, where annealing temperature was gradually reduced by 0.5 °C. This was followed by 35 cycles of 95 °C for 45 s, annealing at 55–63 °C for 45 s and elongation at 72 °C for 30 s and a final elongation step at 72 °C for 5 min. PCR amplification product quality was confirmed via 2% agarose gel electrophoresis.

### 2.7. Pyrosequencing

PCR amplification products were subsequently used for pyrosequencing. Products were first mixed with 40 μL of Binding buffer (Qiagen), 1 μL streptavidin-sepharose beads (GE Healthcare, Buckinghamshire, UK, and 19 μL of water. The mixture was placed on a shaker to allow bead binding to biotin-labelled DNA strands for 15 min. Simultaneously, 25 μL annealing buffer with 0.08 μL sequencing primer mix was dispensed into the Q24 pyrosequencing plate. Following shaking, a Pyromark Q24 Vacuum Workstation (Qiagen) was used to capture and isolate bead-bound PCR products. Following cleaning, biotinylated products were released into the annealing mix prefilled Q24 pyrosequencing plate. The plate was placed on a heater at 80 °C for 2 min and cooled for 3 min at room temperature allowing annealing of sequencing primer to the capturedDNA strands. In the next step, Pyromark Gold Q24 reagents (Qiagen) were carefully added in pyrosequencing dispensation cartridge including the four nucleotides (A, C, G, T), the substrate, and enzyme mix. The enzyme mixture contained DNA polymerase, ATP sulfurylase, luciferase, and apyrase, while the substrate mixture consisted of adenosine 5′ phosphosulfate (APS) and luciferin. The number of nucleotides were determined by PyroMark Q24 software. Negative and positive control samples confirmed consistency of each assay. Following loading the enzyme mix, substrate, and nucleotides into the cartridge, Q24 pyrosequencing plate and cartridge were placed into sequencing machine (Figure 1). Assay design was set up via PyroMark Q24 Software prior to run start. The data from sequencing were analysed by using the PyroMark Q24 Software 2.0.

### 2.8. Statistical Analysis 

SPSS (IBM SPSS Statistics 21) software was used to analyse the results. A one-way analysis of variance (ANOVA) was performed to assess the comparison among the three groups. The threshold *p* < 0.05 was accepted as statistically significant. GraphPad Prism 5 (GraphPad, La Jolla, CA, USA) was performed to analyse the data. Numerical data from three BM-hMSC lines were pooled for analysis purposes. Experimental data are represented as mean values ± standard deviation (SD) and describe a minimum of three independent replicates. 

## 3. Results

### 3.1. Immunophenotype and Differentiation Capacity Are O_2_-Independent

BM-hMSCs, as anticipated, expressed high levels of CD73, CD90, and CD105, while displaying negligible levels of CD14, CD19, CD34, CD45, and HLA-DR across all settings (Appendix A). Furthermore, the first passage of BM-hMSCs was cultured in either adipogenic, chondrogenic, or osteogenic differentiation media in the three different oxygen settings displaying the retention of their differentiation capacity (Appendix A).

### 3.2. Physoxia Upregulates SOX-2 But Not OCT-4 or NANOG 

Physoxia increased approximate 2-fold expression of the SOX-2 (vs. AO) pluripotency marker, but no significant differences were noted for OCT-4 or NANOG gene expression. Relative SOX-2 gene expression levels were increased in 2%WKS (vs. AO) across three independent BM-hMSCs first passage samples (BMA-16 (1.98 ± 0.29, *p* < 0.05), BMA-20 (2.29 ± 0.44, *p* < 0.05), and BMA-25 (1.99 ± 0.28, *p* < 0.05)), and again in 2%PG cultures (BMA-16 (1.45 ± 0.45), BMA-20 (2.17 ± 0.54, *p* < 0.05), and BMA-25 (1.84 ± 0.44, *p* < 0.05)) (Figure 2). 

### 3.3. Global DNA Hypermethylation in Air Oxygen Exposed BM-hMSCs 

Immediately following on from isolation, global 5mC and 5hmC DNA methylation levels were assessed in AO, 2%PG and 2%WKS. A reduced level of global DNA methylation was noted from hMSCs (BMA-16, BMA-20, and BMA-25, first passage) cultured in physoxia conditions in comparison to AO. The percentage of 5mC in total DNA in BMA-16 was significantly reduced in 2%PG (0.67% ± 0.17, *p* < 0.05) and 2%WKS (0.54% ± 0.08, *p* < 0.01) compared to cells cultured in AO (0.99% ± 0.10). This position was reflected with BMA-20 in both physoxia conditions 2%PG and 2%WKS (0.36% ± 0.05, *p* < 0.05; 0.25% ± 0.05, *p* < 0.01, respectively) vs. AO (0.61% ± 0.015). BMA-25 displayed significant reduction in 2%WKS (0.37% ± 0.06) in comparison to those from 2%PG and AO (0.64% ± 0.16 and 0.85% ± 0.06), respectively (Figure 3A). Taken together, BM-MSCs displayed significant reductions in global 5mC levels in 2%WKS (0.39% ± 0.15, *p* < 0.05) when compared to either 2%PG or AO (0.56% ± 0.17 and 0.82% ± 0.2) (Figure 3B).

Significant decreases were also noted in the global 5hmC DNA level for BMA-16 (0.009% ± 0.002, *p* < 0.05) cultured in 2%WKS in comparison to 2%PG or AO (0.016% ± 0.0036; 0.015% ± 0.0008, respectively). Similar reductions were noted with BMA-20 in 2%WKS (0.009% ± 0.0015, *p* < 0.01) in comparison to AO (0.017% ± 0.002). No significant differences were noted for BMA-25, but decreased 5hmC was noted generally in reduced oxygen settings vs. AO (Figure 3C). Taken together, BM-MSCs displayed a significant reduction in global 5hmC levels in 2%WKS (0.009% ± 0.0007, *p* < 0.05) vs. AO (0.019% ± 0.004) (Figure 3D).

### 3.4. Decreased DNMT3B mRNA Associates with Physoxic Culture

The first passage of BM-MSCs were used for gene expression experiments. Maintenance methylase, DNMT1, expression levels were increased in 2%WKS BMA-16 (1.4-fold, *p* < 0.05) when compared to AO. No significant changes in expression were noted for de novo methylase DNMT3A. DNMT3B expression was reduced in BMA-16 and -20 (0.43-fold and 0.38-fold, respectively, *p* < 0.01) cultured in 2%WKS, and BMA-16 and -25 (0.57-fold and 0.6-fold, respectively, *p* < 0.05) exposed 2%PG condition versus AO. DNMT3L was markedly lower in 2%PG and 2%WKS BMA-16 (0.76-fold, *p* < 0.01 and 0.69-fold, *p* < 0.05) in comparison toAO. Furthermore, a decreased expression of DNMT3L in 2%WKS was noted for all (Figure 4). In summary, DNMT3B expression was significantly decreased in both 2%PG and 2%WKS reduced oxygen environment (0.64-fold and 0.59-fold, *p* < 0.05) when compared to AO, while relative expression changes of DNMT1, DNMT3A, and DNMT3L were not significant.

Ten-eleven translocation methylcytosine dioxygenase 1 (TET1) expression was decreased in BMA-16 (0.7-fold, *p* < 0.05, and 0.64-fold, *p* < 0.01) and BMA-20 (0.48-fold and 0.39-fold, *p* < 0.01) in 2%PG and 2%WKS, respectively, compared to AO. BMA-25 TET1 expression in 2%PG (0.6-fold, *p* < 0.05) was significantly less than AO. Overall, a decreased level of TET1 expression was observed in 2%PG and 2%WKS physoxia conditions (0.7-fold and 0.77-fold, *p* < 0.01), respectively, versus AO. There was no significant alteration in gene expression of TET2 and TET3 (Figure 5).

Decreased relative gene expression of HIF1A was noted throughout. BMA-16 in 2%PG (0.76-fold ± 0.11, *p* < 0.05) and 2%WKS (0.76-fold ± 0.10, *p* < 0.05) was reduced when compared to those in AO. We noted reduced HIF1A gene expression in BMA-20 under the 2%WKS (0.59-fold ± 0.09, *p* < 0.01) condition, while BMA-25 in 2%PG (0.66-fold ± 0.18, *p* < 0.05) and 2%WKS (0.59-fold ± 0.15, *p* < 0.05) displayed similar levels of reduced expression versus those in AO (Figure 6). In contrast to above, HIF2A expression was significantly elevated in BMA-16 (2.2-fold ± 0.54, *p* < 0.05) in 2%PG, and BMA-20 in 2%WKS (2.04 ± 0.54, *p* < 0.05). Combined analysis of all BM-hMSCs indicated a significant increase in HIF2A gene expression (1.71-fold ± 0.51 and 1.67-fold ± 0.34, *p* < 0.05) in 2%PG and 2% WKS, respectively (Figure 6). 

### 3.5. DNMT3B and TET1 Protein Expression in BM-hMSCs Cultured in Physiological Normoxia

Transcriptional analysis outcomes were subsequently verified via protein analysis. This confirmed that transcriptional downregulation was accompanied by reduced DNMT3B protein levels in the first passage of BM-hMSCs cultured in reduced oxygen conditions (Figure 7). HIF1A protein was undetectable in BMA-16 and BMA-20. No difference in the level of HIF1A protein was apparent in 2%PG and 2%WKS for BMA-25 (Figure 7). Inspection of all BM-hMSCs suggested increased HIF2A protein levels with 2%PG and 2% WKS (Figure 7).

### 3.6. Reduced Oxygen Increases Methylation of DNMT3B Promoter 

Pyrosequencing was used to determine the methylation of the DNMT3B gene promoter in the first passage of cells. DNMT3B promoter methylation levels were significantly higher in 2%WKS (43%) than in AO (13%) in BMA-16. Similarly, BMA-25 in 2% PG (42%) and 2% WKS (46%) and BMA-20 in 2%PG (38%) and 2%WKS (36%) displayed significant elevation of methylation levels when compared to AO (13% and 7%, respectively). Overall, DNMT3B promoter methylation levels of 34% (2% PG) and 42% (2% WKS) versus 11% (AO) were noted. DNMT1, DNMT3A, and DNMT3L promoter methylation levels displayed no significant change across all three settings (Figure 8).

In contrast to the above, a more heterogeneous pattern of methylation changes was noted for the TET promoters. We noted a significant increase in methylation of TET1 promoter in 2% WKS BMA-16 (13%) and BMA-20 (20%) versus AO (6% and 11%, respectively). Furthermore, methylation level of TET3 promoter was elevated significantly in 2% WKS (20%) when compared to AO (4%) in BMA-25. However, in general, no significant consensus methylation change for the TETs was noted (Figure 9).

## 4. Discussion

Multipotent MSCs have a unique role in tissue regeneration, and thus far, it is clear that the in vitro application of physiological oxygen positively affects cell characteristics including transcription, translation, clonogenicity, growth rates, viability, differentiation, metabolism, and apoptosis [23,24,34,35]. Altered methylation profiles of gene promoters in BM-hMSCs have also correlated with functional changes associated with several metabolic processes, including lipid and fatty acid metabolism, and the regulation of adipogenic differentiation potential during long-term culture (passage 4–12) [36]. Furthermore, DNA methylation at specific promoters during long-term culture of hBM-MSCs (passage 5–15) is suggested to regulate processes including cell development, senescence, proliferation, and genomic stability [36,37]. MSC-based regenerative therapies has some weaknesses, such as poor engraftment and survival following transplantation, inconsistent stem cell potency, genetic and epigenetic instabilities, and premature senescence during ex vivo expansion of bone marrow, adipose, and porcine MSCs (passage 5–10) [38]. Taken together, physoxia and DNA methylation may play a role in BM-hMSCs fate prediction, and potentially provide a system for their selective manipulation, noting that passage number and source may drive variability. Here, we investigated the role of physoxia on global DNA methylation, the transcriptional expression of DNMT/TET enzymes and methylation level of their promoters in hMSCs. We determined that global methylation is oxygen-sensitive and associates with the regulation of DNMT3B transcription and also translation.

MSCs (amniotic tissue, chorionic tissue, liver, umbilical cord, bone marrow-derived, passage 2–8) can be characterised by an immunophenotype with a robust (CD105, CD73, and CD90) negligible (CD45, CD34, CD19, CD3, CD11b, and HLA DR) surface expression of a number of antigens [39,40,41]. Consistent with this, we demonstrated expression of CD73, CD90, and CD105 and a lack CD14, CD19, CD34, CD45, and HLA-DR expression irrespective of the oxygen condition applied. Previous reports have detailed that SOX-2 mRNA levels were increased in hMSCs (dental pulp, passage 5) cultured in low oxygen settings (5% and 3% O_2_) but not Nanog or OCT-4 [42]. Our observations were in broad agreement with this. SOX-2 is expressed in hMSCs (bone marrow, adipose-derived, passage 1–7) at low levels in the early passage and the decrease with increased passage number is potentially linked to maintenance of cell proliferation and multipotency [43,44].

HIF1A is described as an acute responder to reduced oxygen, while HIF2A provides a long-term chronic response in human embryonic stem cells [45]. BM-hMSCs (passage 3) exhibit a reduction in differentiation into adipose and bone tissue, and HIF1A protein upregulation in response to physiologic oxygen tensions [46]. Urine stem cells (USCs), dental pulp stem cells (DPSCs), amniotic fluid stem cells (AFSCs), and BM-MSCs (passage less than 5) cultured under reduced oxygen levels (5% O_2_) showed a significantly increased proliferation rate, an elevated S-phase profile, and higher level of HIF1A gene expression [25]. Furthermore, placenta-derived hMSCs were noted to display upregulation of HIF2A at transcription and translational levels in a 5% O_2_ culture setting [47]. We also noted elevated HIF2A transcript and protein in physoxia but not in HIF1A after long-term culture.

DNA methylation changes are associated with lineage specification during stem cell differentiation [48]. For example, chondrogenic differentiation is associated with DNA hypomethylation at many key cartilage gene loci, such as ACAN and SOX9, and enhancer regions in BM-hMSCs (passage 2–7) [49]. In contrast, global DNA hypomethylation is associated with inhibition of differentiation and increased histone acetylation in hESCs [50]. While global DNA hypomethylation is a feature of ESCs, subsequent differentiation is accompanied by an accumulation of DNA methylation [51] on both locus-specific and global levels [52,53]. 

DNMT1 is described as having a role in the maintenance of self-renewal and differentiation capacity in hMSCs (bone marrow, passage 2–9) [54], but it is unclear what effect physoxia has on this, if any. Increased gene and protein expression of DNMT1 and DNMT3B enzymes in cardiac fibroblasts, suggested to be due to their promoter regions contain a consensus sequence for a HIF1A-binding, and their expression was regulated by HIF1A [55]. We observed an overall increase in DNMT1 gene expression in 2% oxygen and no alterations in DNMT3A or DNMT3L gene expression or promoter methylation in physoxia. Human colorectal-, melanoma-, and neuroblastoma-derived cancer cells cultured in a low oxygen environment (1%) showed decreased global DNA methylation [18,56]. Similarly, we observed a significant reduction in 5mC and 5hmC levels, reduced gene and protein expression of DNMT3B (except BMA-25), and elevated HIF2A in physoxia. DNMT3B specific promoter methylation indicated that physoxia increased promoter methylation levels of DNMT3B. Importantly, the decrease in gene expression of DNMT3B was correlated with an increase in promoter methylation. We have provided the first demonstration that 5mC and 5hmC levels are oxygen-sensitive in BM-MSCs, where de novo methylation is linked to physoxia and correlates directly with transcriptional and translational regulation (except BMA-25) of the de novo methylase DNMT3B. Though compelling, further studies are required before our fundamental biology findings can be translated into an applied setting.

Reduced oxygen conditions (1% O_2_) upregulated TET1, increased global 5hmC, and increased 5hmC level at hypoxia response genes in neuroblastoma cells [56]. Moreover, reduced oxygen (0.5% O_2_) significantly increased the expression of pluripotency-associated OCT4, NANOG transcription factors, TET1, TET3, and global 5hmC level in hESCs [57]. In contrast, we observed a reduction in TET1 gene expression in physoxia in the absence of significant TET1 promoter methylation alterations. 

Epigenetic mechanisms orchestrate MSCs fate, functional homeostasis and multilineage differentiation potential. Increasing evidence suggests that epigenetic mechanisms support MSC-mediated tissue regeneration through cell transplantation or pharmacology-based therapeutics [58]. Manipulation of specific epigenetic marks holds promise for effective application and mimicry of organismal homeostasis for future research. Improved understanding of the epigenetic regulation of MSCs will enhance our knowledge and confidence in regenerative medicine applications. 

## Figures and Tables

**Figure 1 cells-10-01032-f001:**
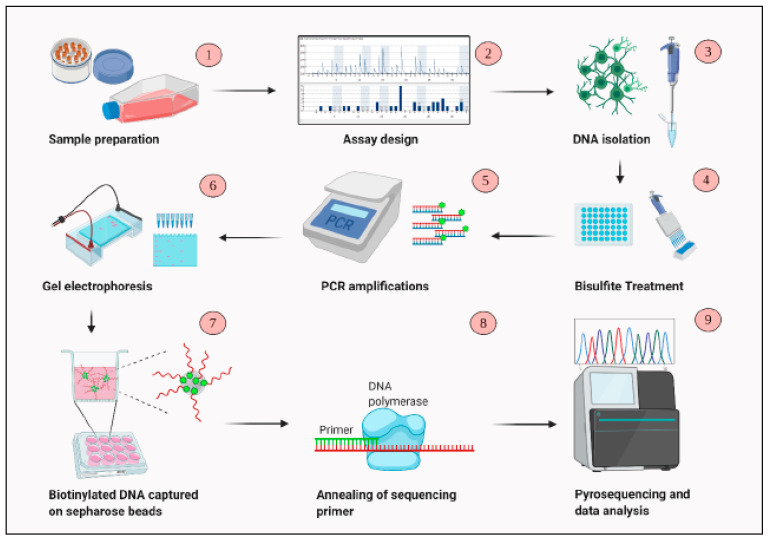
An overview of the steps involved in DNA methylation analysis using pyrosequencing.

**Figure 2 cells-10-01032-f002:**
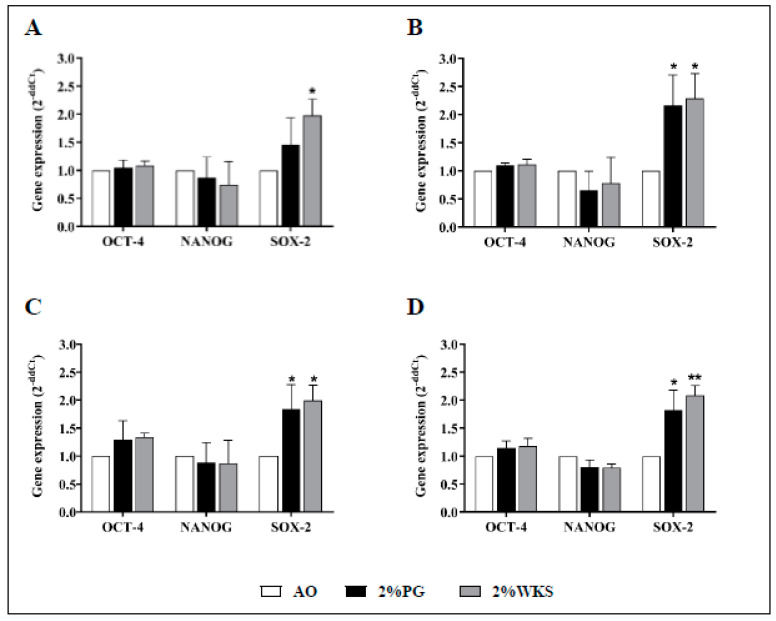
Physoxia upregulation of SOX2. (**A**) hMSC-1, BMA-16; (**B**) hMSC-2, BMA-20; (**C**) hMSC-3, BMA-25; (**D**) hMSC-profile, BM-MSCs (BMA-16, -20, and -25). Gene expression results normalised to the β-actin gene expression. X-axis represents three pluripotency markers (NANOG, OCT-4, SOX-2) and Y-axis indicates fold change of physoxia vs. AO hMSCs. Fold change calculated via 2^−ΔΔCT^ methodology. The values are relative mean (*n* = 3), * *p* < 0.05, ** *p* < 0.01, standard deviation (SD) is indicated by error bar.

**Figure 3 cells-10-01032-f003:**
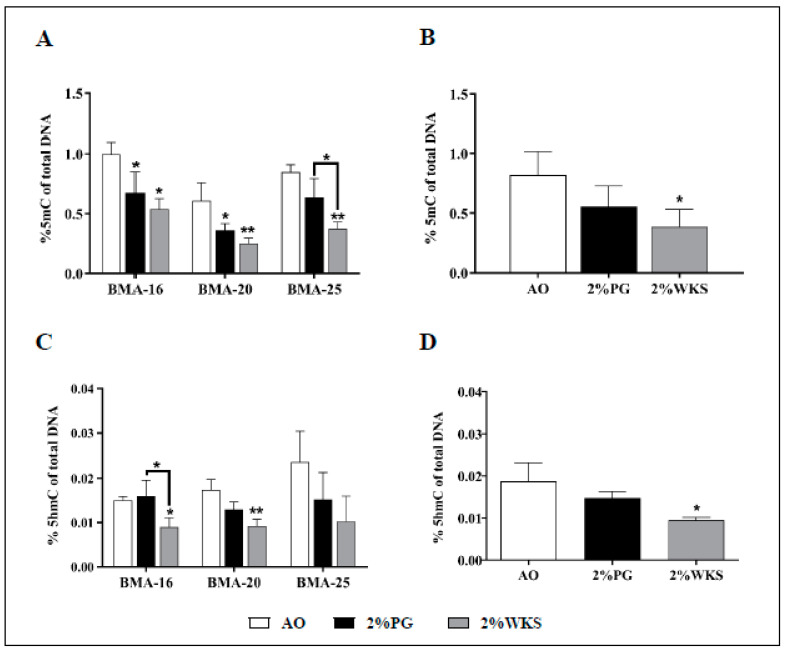
Air oxygen driven upregulation of 5mC and 5hmC global levels. (**A**) and (**B**) %5mC, (**C**) and (**D**) %5hmC. Three BM-hMSCs were incubated in AO and physoxic conditions (2%PG and 2%WKS). MethylFlashTM Methylated DNA quantification kit and MethylFlashTM hydroxymethylated DNA quantification kit were used to measure the methylated DNA. Y-axis shows the absorbance value (450 nm). X-axis represents three different oxygen culture conditions. The values are relative mean (*n* = 3), * *p* < 0.05, ** *p* < 0.01 vs. AO, and standard deviation (SD) is indicated by error bar.

**Figure 4 cells-10-01032-f004:**
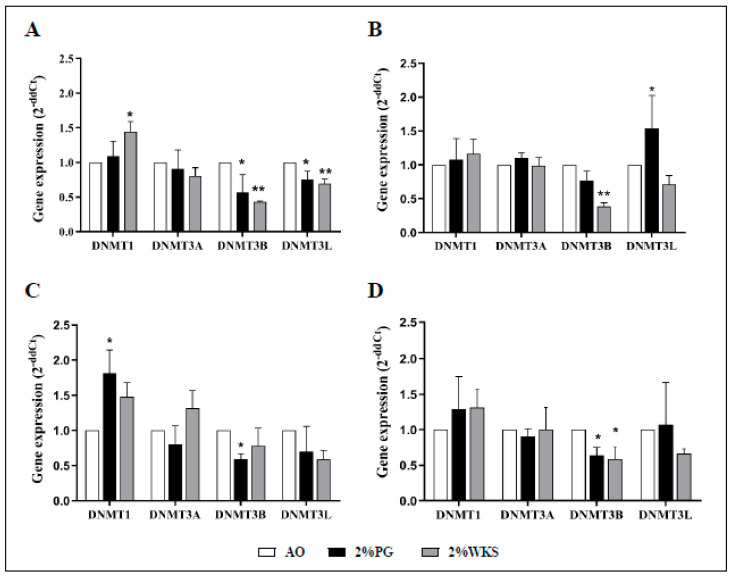
DNMT3B is transcriptionally downregulated in physoxia. (**A**) hMSC-1, BMA-16; (**B**) hMSC-2, BMA-20; (**C**) hMSC-3, BMA-25; (**D**) hMSC-profile, BM-MSCs (BMA-16, -20, and -25). Gene expression of DNMTs was performed with BM-hMSCs following isolation in AO and physoxia settings. β-actin gene expression was used to normalise the gene expression results. X-axis represents four DNMTs enzymes and Y-axis indicates fold change of physoxia vs. AO. Fold change calculated via 2^−ΔΔCT^ methodology. The values are relative mean (*n* = 3), * *p* < 0.05, ** *p* < 0.01 vs. AO, and standard deviation (SD) is indicated by error bar.

**Figure 5 cells-10-01032-f005:**
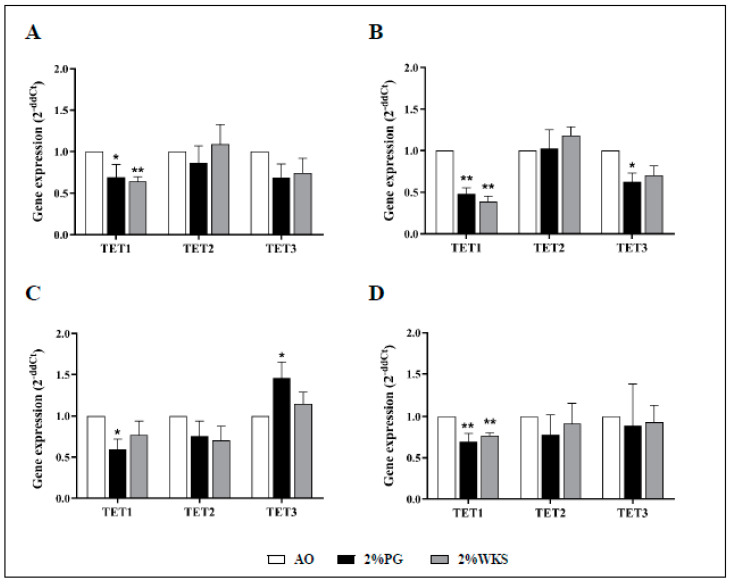
TET1 is downregulated in physoxia cultured hMSCs. (**A**) hMSC-1, BMA-16; (**B**) hMSC-2, BMA-20; (**C**) hMSC-3, BMA-25; (**D**) hMSC-profile, BM-MSCs (BMA-16, -20, and -25). Gene expression results normalised to the β-actin gene expression. X-axis represents three TET enzymes and Y-axis indicates fold changes of physoxia vs. AO hMSCs. Fold change calculated via 2^−ΔΔCT^ methodology. The values are relative mean (*n* = 3), * *p* < 0.05, ** *p* < 0.01 vs. AO, and standard deviation (SD) is indicated by error bar.

**Figure 6 cells-10-01032-f006:**
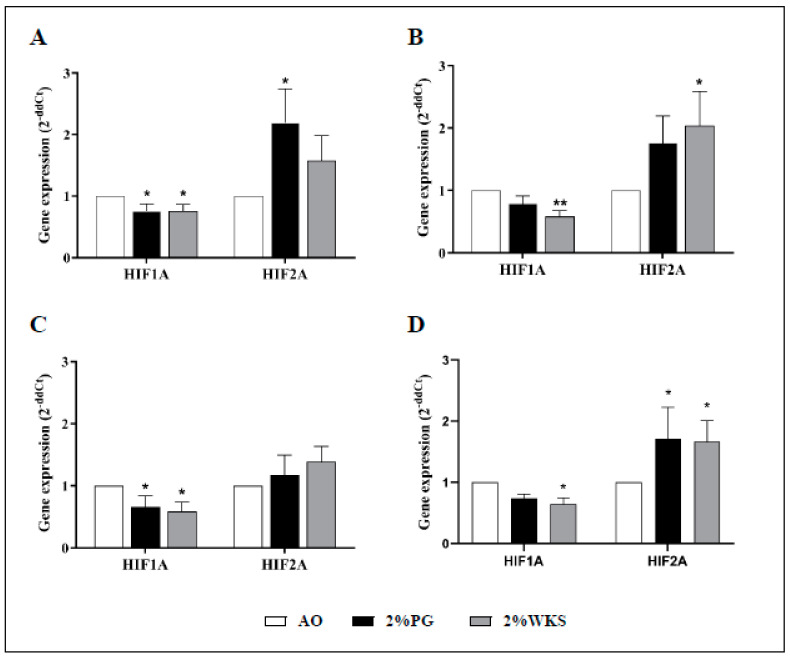
HIF1A and HIF2A are differentially expressed in physoxia. (**A**) hMSC-1, BMA-16; (**B**) hMSC-2, BMA-20; (**C**) hMSC-3, BMA-25; (**D**) hMSC-profile, BM-MSCs (BMA-16, -20, and -25). Gene expression results normalised to the β-actin gene expression. X-axis represents HIFs enzymes and Y-axis indicates fold changes of physioxia vs. AO cultured hMSCs. Fold change calculated via 2^−ΔΔCT^ methodology. The values are relative mean (*n* = 3), * *p* < 0.05, ** *p* < 0.01 vs. AO, and standard deviation (SD) is indicated by error bar.

**Figure 7 cells-10-01032-f007:**
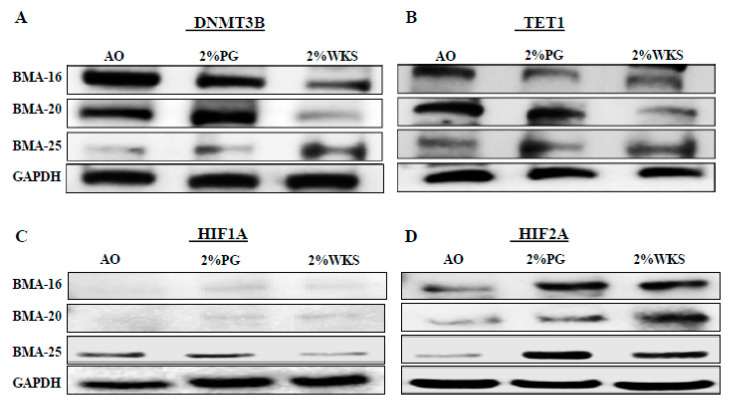
DNMT3B, TET1, and HIF1A, but not HIF2A, are translationally downregulated in physoxia. (**A**) DNMT3B, (**B**) TET1, (**C**) HIF1A, and (**D**) HIF2A protein expression levels in BM-hMSCs in three conditions. Protein was isolated from three BM-hMSCs at day 21 following incubation in AO and physoxia (2%PG and 2%WKS) conditions. GAPDH included for control.

**Figure 8 cells-10-01032-f008:**
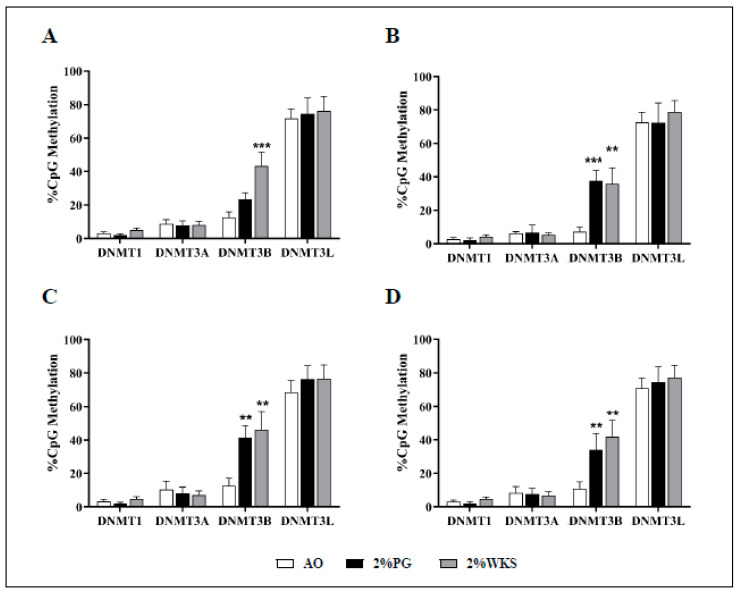
Air oxygen reduces DNMT3B promoter methylation. A set of CpG islands in DNMT gene promoters were evaluated using pyrosequencing. (**A**) hMSC-1, BMA-16; (**B**) hMSC-2, BMA-20; (**C**) hMSC-3, BMA-25; (**D**) hMSC-profile, BM-MSCs (BMA-16, -20, and -25). Y-axis indicates DNA methylation level (%) at promoter regions. X-axis represents DNMT enzymes. The values are percentage (*n* = 3), * *p* < 0.05, ** *p* < 0.01, *** *p* < 0.001 vs. AO, and standard deviation (SD) is indicated by error bar.

**Figure 9 cells-10-01032-f009:**
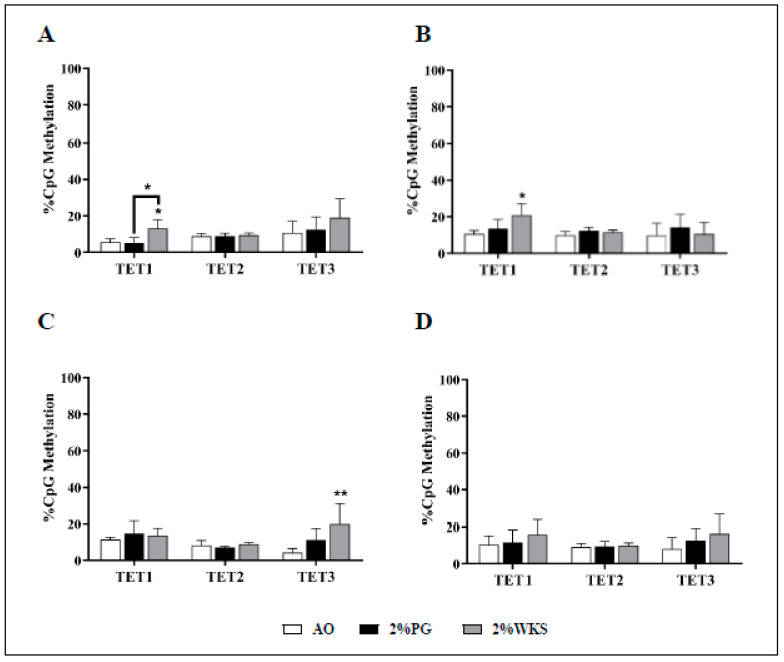
TET promoter methylation is oxygen-independent. A set of CpG islands in TET gene promoters were evaluated using pyrosequencing. (**A**) hMSC-1, BMA-16; (**B**) hMSC-2, BMA-20; (**C**) hMSC-3, BMA-25; (**D**) hMSC-profile, BM-MSCs (BMA-16, -20, and -25). Y-axis indicates DNA methylation levels (%) at promoter regions. X-axis indicates DNMT enzymes. The values are percentage (*n* = 3), * *p* < 0.05, ** *p* < 0.01 vs. AO, and standard deviation (SD) is indicated by error bar.

## Data Availability

Data available in a publicly accessible repository.

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
