# Peer review of "DNMT3B Is an Oxygen-Sensitive De Novo Methylase in Human Mesenchymal Stem Cells"

_cells, 2021, doi:10.3390/cells10051032_

Round 1

Reviewer 1 Report

The BMA-25 sample behaves quite differently from the other two as seen in Fig. 7. The conclusions reached by the authors are not objective in light of the results of the experiments with each of the samples. There is too much variability between samples and I think the number of samples should be increased. 

Minor Concerns:

1-Please, correct pyhsoxia on page 2, line 27 of the manuscript. 

2-A comma is missing between references 10 and 11 on page 3 line 52. 

Author Response

We thank Reviewer 1 for their constructive feedback. We have detailed our responses, point by point, and the manner in which these have been addressed in attached document.

Reviewer 2 Report

Interesting paper highlighting the role of oxygen on epigenetic regulation of hMSC.

The weakness is that the experiments are done only on three MSC culture (n=3) but the results are consistent

The experiments are clearly described and well presented.

The global goal of the paper is reached by the different tests performed.

I think that the paper can be published in his current form.

Author Response

We thank Reviewer 2 for their constructive feedback. 

Reviewer 3 Report

The manuscript entitled "DNMT3B is an oxygen-sensitive de novo methylase in human mesenchymal stem cells" is an experimental manuscript that discussed the effects of the oxygen levels on the proliferation and differentiation of human derive mesenchymal stem cells (MSCs) as well as the effects of methylation and its impact on some of the genes. Though this is an interesting concept, there are multiple major issues in the manuscript as outlined below.

Introduction:

  1. Please mention about the source and the passage number of human MSCs used in this study in the abstract
  2. Introduction needs grammar check and spelling errors
  3. Page 3 line 43 - "differentiation into replacement cell types" is a very strong statement, since the MSCs do not differentiate to replace all the cell types, they are multipotent in nature.
  4. More background information about the oxygen concept and epigenetics should be discussed. When using a term for the first time, please explain the term with abbreviation.
  5. The abbreviation of the words should be in parentheses (page 3 line 54 and 55)
  6. The introduction of oxygen levels and cancer is deviating from the topic which distracts the readers.
  7. Please mention in vitro in italics, liter should be L not l.
  8. When mentioning about previous studies, mention about the source and the passage number of MSCs used in the study, as these are the major factor that influences the characteristics of MSCs
  9. At the end of the introduction, please mention the goal of the study.

Materials and methods:

  1. Even though the MSCs passage and plating was adapted from previous study, please mention the methods briefly
  2. It is not clear from the methods if the 3 human cell lines were pooled or not
  3. For every analysis mentioned in the methods section, please mention the passage number used.
  4. Under flow cytometer section, please mention the concentration of the antibodies used.
  5. Please give the components of the FACS media used for the analysis (page 6 line 118)
  6. Similarly, what buffer was used to dilute the antibodies (Page 6, line 116)
  7. Why different amounts of DNA were added to for the 5mc and 5hmc analysis? Would this not this interfere with the result comparison?

Results:

  1. Please mention the passage number used for each of the result section
  2. For each of the graph, please mention in the graph as to which of the 3 human MSCs line was used. This will make it easier for the readers to follow.
  3. As the results for each cell line is different, how can this be used for translational purposes? please mention the impact of these.
  4. If the results are different for different cell lines, would it not be beneficial to pool different cell line to determine the effects of oxygen levels on the cell lines?

Discussion:

  1. Page 20, line 325-329 - too strong conclusions, these results might be different based on source and the passage number of MSCs, also the results need not be the same for different cell lines.

Overall:

The manuscript lacks details in methods section and there are multiple points that needs clarification. Please mention the application of the study. Those issues should be remedied if possible but, in this form, this manuscript should be rejected.

Author Response

We thank Reviewer 3 for their constructive feedback. We have detailed below our responses, point by point, and the manner in which these have been addressed.

Round 2

Reviewer 1 Report

The authors anticipated variability between samples, as they answered is consistent with most multiple cell line analysis papers. However, they did not added any references to demonstrate it. 
Afterwards, the authors believe BMA-25 is not overall different from other cells. However, the results by western presented are indicating the opposite.

Minor concerns:

Page 8, line 169, Nanodrop is misspelling in the manuscript.

Supplementary Information 3, Alcian Blue is misspelling in the figure.

Reviewer 3 Report

Authors were able to answer and clarify all the points raised in the previous review and convince this reviewer with the revised manuscript. Thank you for the satisfactory answers, there are some typos still present in the manuscript, for example:

Pg 4 – line 78 – typo in MSCs (it is written as MSCc)

Please address these typos.